# Perceptions of Quality of Life during the Pandemic: A Case Study on B40 Single Mothers

**DOI:** 10.3390/ijerph191912219

**Published:** 2022-09-27

**Authors:** Siti Marziah Zakaria, Norehan Abdullah, Noremy Md. Akhir, Aizan Sofia Amin, Asila Nur Adlynd Mohd Shukry, Mohd Radzniwan Abdul Rashid, Wan Nurdiyana Wan Yusof

**Affiliations:** 1Centre for Research in Psychology and Human Well-Being, Faculty of Social Sciences and Humanities, Universiti Kebangsaan Malaysia, Bangi 43600, Selangor, Malaysia; 2School of Economics, Finance and Banking, College of Business, Universiti Utara Malaysia, Sintok 06010, Kedah, Malaysia; 3Faculty of Medicine and Health Science, Universiti Sains Islam Malaysia, Nilai 71800, Negeri Sembilan, Malaysia

**Keywords:** single mothers, quality of life, B40, pandemic, COVID-19, MCO

## Abstract

During the pandemic, the lives of B40 single mothers were severely affected, especially in terms of social, economic, and psychological factors. The reduction of income caused by the crisis has forced single mothers and their children to live frugally and without luxury. They also had to perform more than one job at a time during the pandemic to meet their children’s needs. A qualitative study has been conducted to explore the perceptions of quality of life during the pandemic among B40 single mothers. Respondents were selected based on demographic characteristics established before the study. A focus group discussion has been conducted on ten (10) B40 single mothers in Balik Pulau, Penang. Single mothers were selected based on purposive sampling. They had to meet the inclusive criteria required to participate in the study, specifically: aged between 20 and 60 years old, belonged to the B40 income category, had children under 18 years old living together, and became single mothers due to divorce or death of husbands. The focused group discussion explored issues related to qualities of life during the pandemic. B40 single mothers expressed their concern about access to health facilities, security in residence, children’s education, and economic stability. These areas of life quality have been significantly affected especially during Movement Control Order (MCO). In short, the quality of life of these single mothers has been significantly affected by the pandemic. Their vulnerability towards stress, anxiety, and depression have worsened due to financial issues. Besides the need for emotional and social support, this study found that these single mothers entail financial support.

## 1. Introduction

The COVID-19 pandemic that began at the end of 2019 had significant impacts on all aspects of life globally, including in Malaysia. The implementation of Movement Control Order (MCO) starting on 18 March 2020, to curb the spread of the pandemic, had a profound effect on all levels of society regardless of age, religion, and race due to the sudden changes in daily lives and mobility. The cultivation of new norms in the community implemented by the government also had psychological impacts on people, causing many to feel scared, depressed, anxious, and worried about being infected with COVID-19. B40 single mothers also said they were constantly anxious, fearful, and worried their family members might be infected with COVID-19. A previous study also found that emotions such as fear, anxiety, and worry can be triggered by the COVID-19 outbreak [1]. Some aspects of quality of life such as mobility, the ability to perform daily activities, resting, and the quality of sleep have been negatively affected during the COVID-19 pandemic [2].

### The COVID-19 Pandemic: Challenges Faced by B40 Single Mothers

During the pandemic, the lives of B40 single mothers were severely affected, especially in terms of social, economic, and psychological factors. The reduction of income caused by the crisis forced single mothers and their children to live frugally and without luxury. They also had to perform more than one job at a time during the pandemic to meet their children’s needs. Some jobs they had were conducting business online, selling fruit, cleaning houses, and performing food delivery. Unfortunately, some single mothers found themselves in a desperate situation and had to quit their jobs to care for their young children. Those single mothers had to make that decision since they could not afford the cost of childcare due to the financial problems they had faced. Consequently, stress caused by several factors such as job loss, financial instability, loneliness, and lack of food can aggravate mental health problems [3,4].

Therefore, most B40 single mothers experienced stress, anxiety, and depression during the COVID-19 pandemic. Among the reasons that made them feel depressed were financial problems, lack of social support, and concern for the safety of themselves and their children. Financial difficulties are the main problem faced by most B40 single mothers in Malaysia. They had already experienced the same situation before the pandemic crisis, and it worsened during the pandemic because some of them lost or had to quit their jobs, or their businesses were adversely affected. These situations made it difficult for them to meet the family needs and, at some point, triggered pressure on them. Similarly, the researchers noted that the worse the economic hardship faced by single mothers, the higher the likelihood of them experiencing mental health problems [5]. Thus, single mothers may be unable to perform their daily tasks and go to work due to their mental health problems.

The implementation of movement control order during the pandemic has prevented single mothers from meeting family and friends and participating in assistance programs. The researchers stated that physical distancing has adversely affected individuals’ mental health, thus causing them to experience post-traumatic stress disorder (PTSD), anxiety and obsessive–compulsive disorder [6]. As a result, single mothers did not get social support from people around them. A past study also found that single mothers are at risk of experiencing mental health problems due to high stress levels and lack of social and physical support [7]. Single mothers who have a lack of social support will have less effective parenting behaviors [8,9]. On the other hand, single mothers with strong social support will have a higher confidence level in performing as parents and have better relationships with their children [9]. Thus, this article will explore perceptions of the quality of life of B40 single mothers during the COVID-19 pandemic crisis.

## 2. Materials and Methods

### 2.1. Research Design

The study was conducted qualitatively using phenomenology approach as the research design and the focused group discussion (FGD) on a group of selected single mothers as technique of data collection. A phenomenological approach was used to explore experience and quality of life among single mothers during the COVID-19 pandemic. Focused group discussion is a qualitative approach often used to gain a clear and in-depth understanding of social and psychological issues. Focused group discussions involve small group interviews with several respondents who have the same characteristics (homogeneous) to discuss specific issues by emphasizing the interaction between members in the group where the researcher acts as a moderator. Basically, the purpose of an FGD is to gain an understanding of a particular issue from the perspective and interpretation of a selected group of individuals. Respondents were selected based on inclusion and exclusion criteria established before the FGD.

### 2.2. Research Sample

Single mothers were selected based on purposive sampling. They had to meet the inclusion criteria required to participate in the study, specifically: aged between 20 and 60 years old, belonged to the B40 income category, had children under 18 years old living together, and became single mothers due to divorce or death of husbands. According to the Department of Statistics Malaysia, B40 is referred to a total household income of RM4360.00 and below.

Meanwhile, the exclusion criteria for this study are aged below 20 and over 60 years old, not belong to the B40 income category, had children over 18 years old, and not a single mother. The respondents participated voluntarily and signed the informed consent forms before engaging in focused group discussions.

### 2.3. Data Collection Procedure

Selected single mothers were required to complete demographic forms before the focused group discussion (FGD) began. They were informed about the study ethics and the confidentiality of data and conversations during the FGD. Twelve (12) semi-structured questions were used as FGD guides. The FGD session lasted two (2) hours and was divided into two (2) groups to facilitate the process. All respondents had ample opportunity to participate and speak up because it was conducted in small groups of five (5) respondents each. The FGD ended with a closing and appreciation speech from the researchers.

### 2.4. Research Instruments

The main research instrument used for the FGD was a set of semi-structured interviews which was developed in Malay language. As the FGD aimed to explore the perceptions of quality of life during the pandemic among B40 single mothers, the content of the interview protocol was designed based on the emergence theme during the brain storming stage by the research team and guided by Seligman’s PERMA Model. The domains that were included in the interview were mental health, quality of life, and well-being of single mothers during the COVID-19 pandemic.

The semi-structured interview consisted of two parts, which were the introduction and the key question. A total of 15 questions was developed to explore mental health, quality of life, and well-being of single mothers during the pandemic. The interview protocol was distributed to four panels of subject matter experts comprised of family medicine specialists, psychologists, and economic experts to assess content validity of the interview protocol. They were requested to give comments and to rate the suitability and clarity of each question (yes or no response). Most commented that the questions were relevant, suitable, and clear. After taking into consideration the views and remarks from expert panel members, a final set of questions was constructed, consisting of 12 questions (introduction and key question). Nevertheless, to fulfil the objective of this article, only questions related to the domain of quality of life among B40 single mothers will be highlighted.

### 2.5. Data Analysis

Data were analyzed using the thematic analysis technique to extract themes from the text or data. Researchers have carefully examined the data based on the transcripts to identify recurring patterns. Once identified, themes would be analyzed and interpreted inductively. Then, the researchers would extract the theme of single mothers’ quality of life from the analyzed text.

## 3. Results and Discussion

### 3.1. Background of the Respondents

Ten respondents were interviewed to explore their quality of life during the COVID-19 pandemic. The respondents came from various backgrounds, between 35 and 52 years old. Eight women became single mothers due to the death of their husbands, while the other two were divorcees. The number of children under the care of the respondents interviewed in the FGD ranged from one to nine. The quality of life of respondents with many children was critical during the pandemic. Most respondents had children who were still in primary and secondary schools. They had become single mothers within eight months to 13 years. Most of the respondents worked as entrepreneurs, cleaners, childcare providers, sales agents, and pre-school teachers.

### 3.2. Perceptions of Quality of Life during the COVID-19 Pandemic

This focused group discussion has explored issues related to qualities of life during the pandemic. B40 single mothers expressed their concern about access to health facilities, security in residence, children’s education, and economic stability. These areas of life quality had been significantly affected especially during MCO. The respondents also voiced feelings of helplessness, anxiety, and stress in coping with these issues.

#### 3.2.1. Access to Health Facilities

The government is responsible for providing health services to improve the quality of life of the population in the community. Public health services are the services provided by the government to the people [10]. However, rural communities often experience obstacles and challenges regarding health accessibility [11]. This matter was also stated by the B40 single mothers living in rural areas. They faced several issues regarding health facilities, such as transportation problems and the quality of treatment received in public and private hospitals. A respondent agreed that the lack of comfortable and suitable transportation made it difficult for her to bring a family member to the hospital for treatment:

“It’s hard to take him back and forth to the hospital… It’s difficult to get transportation because I don’t know how to drive, I rode the motorcycle, left it and then we took the bus.”

B40 single mothers have experienced transportation problems as most of them did not possess driving licenses and vehicles that enabled them to travel and to seek treatment at hospitals or clinics. This situation became worse when public transport such as buses and taxis rarely came by when they needed to seek treatment. Ambulance services provided by some hospitals were also too slow. As a result, the treatment could not be administered as soon as possible, thus endangering the patients’ lives. In this situation, single mothers preferred to seek private health services nearby their homes, especially when dealing with emergency cases, despite having to bear the higher treatment costs. This issue was acknowledged by a respondent:

“When it is an emergency, I will go to a private hospital or clinic.”

Furthermore, several B40 single mothers expressed dissatisfaction with the quality of treatment and the long waiting period they had faced in public health service facilities. According to them, the quality of service they received was poor because there had been some issues while getting treatment, such as the time spent in the process of taking blood samples being too long and inefficient. Such issues have bothered single mothers and put them at risk every time they seek treatment from government hospitals. The inefficiency of those health workers can jeopardize the safety of B40 single mothers and other individuals. This situation has been ongoing since before the COVID-19 pandemic. People have also raised their dissatisfaction with the long waiting time for getting treatment. Therefore, the Malaysian Ministry of Health has recommended that the waiting time is stipulated at less than 30 min in all government hospitals and clinics. Yet, the opposite situation occurred in rural public health facilities, where patients had to wait almost 4 h before getting treatment. A respondent shared her opinion and experience in seeking treatment at a public health facility:

“It’s a long wait at the hospital.”

The low status of some individuals may affect the quality of health services they receive. There is a perception that says, since poor and less fortunate individuals are exempted from paying for health services, they deserve to get second-rate quality of service [12]. Some B40 single mothers have also questioned their eligibility for quality treatment services even though they only have to pay RM1 in public health facilities. This situation indicates that health equity still fails to be implemented in providing health services to the rural population. Research has shown that health equity has been linked to social justice, where health care of satisfactory quality has been provided to every individual regardless of socio-economic status [13]. Thus, the health gap needs to be reduced by offering equal health accessibility in urban and rural areas to achieve good health equity [13].

Some single mothers who went to private clinics also said that it took some time before getting treatment because there were too many other patients queuing to apply for medical certificates (MC). As a result, those who were genuinely sick had to queue up too. They also had to wait too long for the treatment. This problem was attributed to insufficient registration counters. Respondent 4 shared her experience of seeking treatment for a child with bone problems at a government hospital. According to respondent, it took years for her child to get a diagnosis, thus depriving him of getting proper treatment.

“My son has bone problems. It took two years to get the result and the approval to insert iron in his spine. During the 2-year pandemic, we could not come for check-ups, and the specialist said the screw has turned loose, and it stuck to the meat.”

In addition, private clinics have prescribed drugs not needed by B40 single mothers and charged them exorbitant additional fees. They were worried because they did not know whether they needed those drugs. They did not have the same knowledge possessed by certified doctors. Therefore, they had to accept and pay for those unnecessary medications prescribed by the clinics. Since they belong to the low-income group, the high cost of treatment is burdensome. This is because their monthly income is just barely enough to cover household expenses and the cost of raising their children. This situation has forced low-income single mothers to do more than one job or work overtime to generate additional sources of income. Respondents shared their experience of working overtime to pay for her children’s expensive medical treatment:

“I am facing two trials. My husband (passed away) due to a heart attack and my son is sick.”

“Yes, the (medical) costs are expensive.”

“Sometimes, other people see me working overtime (OT), like I do not get any aid, so they say, but they don’t know that the treatment cost at the clinic is very expensive, moreover my child is sick, even RM500 is not enough for him alone.”

#### 3.2.2. Security in Residence

B40 single mothers living in the rural areas explained that they had faced several security issues. One of the problems they shared was infrequent police patrols in the villages. Single mothers living in the villages or rural areas feel less safe because there are fewer police patrols in residential areas to monitor suspicious activities that disturb the peace. A previous study also argued that incomprehensive and infrequent police patrols could lead to criminal cases such as burglary in residential areas [14]. This situation has made it difficult for single mothers to seek immediate security assistance as they have to go to the police station. This situation was acknowledged by one of respondents, who stated that the lack of police patrols in certain villages caused her to feel worried about the safety of herself and her family members.

“Depends on the village, there are (police patrols), but not enough.”

Some criminals escaped since the police arrived late at the crime scene. For example, when a disturbance occurs in a flat, the police will only go there after someone has lodged a report. It takes a long time for them to arrive at the scene. This problem has indirectly allowed the offenders to escape, and the same situation will happen again and continue to disrupt the peace of the housing area. In addition, cases of theft in flats located in rural areas are also on the rise, and some cases still have not been resolved. Insufficient police patrols in those areas led to an increase in theft cases, especially during the COVID-19 pandemic, as many individuals were in a desperate situation, thus forcing them to steal for survival. Respondents shared the differences in the level of security in a village and a flat:

“Houses in the village lacked security because many people coming in and out of the area.”

“Other ladies living in villages may feel quite unsafe. I feel safer living in a flat.”

Furthermore, single mothers have also expressed dissatisfaction over the security control by the police as there are still many reported theft cases unsolved. Indirectly, this situation has made those thieves feel they were untouchable and because the police did not arrest them, they kept repeating the same criminal act. Those single mothers also said they were tired of lodging police reports because no action has been taken, despite having filed many reports. This issue threatens the safety of single mothers living with young children, thus affecting their level of trust in the authority. The research showed that index crimes such as property crimes have been widely reported in Malaysia [15].

The increase in such cases indicated the preventive measures undertaken by the authorities failed to curb crime. Even city planning cannot create a safe environment for the community [16]. As stressed by a past researcher, the problem of crime is a big issue that worries the community, but there are still no perfect preventive measures to address that problem holistically [17]. In so doing, the government has sought to increase the national security level by reducing criminal cases in the community. However, those crime prevention measures need to be done holistically with the cooperation of all parties, including government agencies, the private sector, NGOs and the community. Such cooperation can be implemented if the community fully trusts the government in ensuring their safety [18]. This notion has been supported by previous research which argued that the joint involvement of the police and the local community is essential in ensuring safety and public order in a city [17].

#### 3.2.3. Children’s Education

During the COVID-19 pandemic, face-to-face learning sessions could not be conducted due to measures taken to curb the spread of the virus. Therefore, the government has taken the latest initiative by implementing the Teaching and Learning at Home (PdPR) method online or offline. The objective was to ensure the continuation of educational activities; so that the students’ learning process was not affected. Researchers agreed that online teaching and learning played an important role in the Malaysian education system [19]. However, the implementation of PdPR online has posed several challenges for teachers and students [20]. PdPR requires students and teachers to have the necessary electronic gadgets and high-speed internet access while undergoing the teaching and learning sessions. Parents, especially low-income single mothers, faced challenges and pressure to provide those necessities such as electronic gadgets and internet access. Some respondents admitted to feeling depressed during the implementation of PdPR:

“It was stressful during the PdPR.”

PdPR implementation has made life difficult for low-income single mothers because they must provide internet access and gadgets for their school-going children. This situation agrees with a previous study that explained the challenges, such as the provision of materials, knowledge, facilities, skills, and family management, faced by parents during the implementation of PdPR [21]. Researchers also explained that providing gadgets and internet access for children’s learning process has become a burden to low-income families or families with many children [22]. This fact was acknowledged by the respondent, who stated that the inadequacy of gadgets for her children’s learning process caused anxiety in single mother like her. This situation forced them to seek help from others to ensure that their children did not fall behind in their studies, even if they could not afford to buy new gadgets.

“Not enough gadgets for the kids. I had to ask my child to borrow one from other people. Luckily my mother helped us.”

This finding is in line with the findings of a past study that stated there were two main obstacles to conducting PdPR; namely, the ability of parents to provide gadgets and internet access was either weak or unstable [23]. According to what has been shared by B40 single mothers, the preparation for the learning process, such as providing enough gadgets and high-speed internet data, was quite burdensome for them as it required huge expenses. This situation was acknowledged by the respondents:

“Feeling stressed out because need money to buy the reload.”

“When they told me they had to reload, I already knew it in several days. So, I had to think about it.”

The rise in the cost of gadgets caused by high demand during the pandemic has also made it difficult for B40 single mothers to meet the needs in their children’s learning process, thus making them worried about its effects on their children’s education. Insufficient gadgets have caused their children to miss most online learning sessions because they had to share their gadgets with their siblings, who were also still in school. This fact was also reported by a past study that stated most students had to share mobile phones with parents and other siblings for learning purposes [24]. As a result, most children found it hard to concentrate and lost interest in learning during the COVID-19 pandemic. This issue indirectly forced single mothers to find sufficient income to buy gadgets for their children, so they could still follow the learning process through the PdPR method.

Moreover, B40 single mothers also said they had difficulty accessing high-speed internet for their children, as the areas where they lived had network problems. As a consequence, this issue disrupted their children’s online learning process. Similarly, researchers stated that low internet access could thwart the proper implementation of PdPR [22]. This problem could also affect schoolchildren because it would be difficult for them to access reference materials needed to complete school assignments from the internet. The depletion of internet data has also disrupted the interaction between students and teachers, and it has caused them to lose focus on learning and to eventually, give up. Therefore, PdPR sessions would be incomplete if any of those factors could not be provided sufficiently. Respondents commented on the difficulties related to internet access faced by their children during the PdPR process:

“Now we are using Unifi too because it will be costly to purchase data.”

“The Internet is slow, depending on the location, some places are fast, but others are slow.”

“But not all telcos provide high-speed Internet, I must get a new line. Now, I have too many phone numbers.”

“The data quota exhausts so quick, the Internet is slow; more money has to be spent if we want to renew data subscription, so we can get faster Internet access.”

B40 single mothers also said that the effectiveness level of PdPR among their children was quite unsatisfactory as most of them missed some sessions, thus causing them to fall behind in their learning. Therefore, it is vital for children sitting for important examinations, such as PT3, SPM, and STPM, to take additional classes to re-learn the syllabus they have missed. Respondents also expressed their frustration over their children’s falling behind and their sadness at not being able to help their children academically:

“Now they have a different syllabus, not the same as we had previously; we cannot help them because we don’t know.”

“School days have just gone like that; suddenly, he is in Year 6.”

The implementation of the new norm of the learning system, to some extent, has an impact on the socio-economic and emotional aspects of the B40 single mothers. That has been the case because the high cost of living during the pandemic is not in line with their household incomes, still categorized below the poverty line. This situation had placed this group permanently under the poverty line even before the COVID-19 pandemic began.

#### 3.2.4. Economic Stability

Economic stability is crucial to the single mothers’ lives and their children. According to them, their lives would improve if they could achieve economic stability so they can get better quality services in healthcare, safety, and education for their families. The sudden loss of husbands as the heads of the families and sole breadwinners due to death caused the lives of single mothers to fall into uncertainty and made them lose their source of income:

“The misery was overwhelming within eight months after he left us. The loss of income was substantial.”

“Soon after his death, I felt so stressed because I had lost the main source of income. My husband was in the army, so it was quite a hassle for me to withdraw his pension.”

The lives of single mothers who are often affected due to financial problems have prompted them to work overtime to meet the needs of the family because they are the sole breadwinners.

“Sometimes people see me working overtime (OT) as if I don’t get any aid, but they don’t know that the clinic bills are very costly…”

This situation has affected the relationship with their children because B40 single mothers became too tired and did not have enough time to spend with them, thus causing the children to feel neglected. Respondents admitted that it was difficult for them to balance their commitments between careers and children as they had to balance their roles as sole breadwinners and primary caregivers of their families. Respondent also admitted that she had to quit her job because it was difficult to manage her time.

“Single mothers can work, but there are limitations.”

“I have tried working part-time, but it was difficult because there was no one to look after my child.”

“After having the fourth child, I quit my job at the factory because I could not allocate the time between taking care of the kids and work.”

According to those single mothers, children are meaningful gifts from God because they motivate them to go on with their lives. Researchers have stated that there are three ways where the relationship between parent and child affects the life of a single mother: (i) parents view their children as the utmost motivation in making positive changes, (ii) parents prioritize the needs of their children, and (iii) parents consider their children a motivating factor in getting themselves out of poverty [25,26]. B40 single mothers also said their children motivated them to strive for prosperity and financial stability. Parenthood has made them aware of the responsibility to be good role models for their children [27,28]. They also explained the importance of children as a motivating factor in achieving long-term goals of improving their standard of living [29]. This factor encourages single mothers to prioritize the happiness of their children over their own. This situation was acknowledged by the respondents:

“The kids come first; they have the priority when it comes to buying things. They are the reason why I go to work.”

“But when it comes to buying clothes, I don’t mind getting the cheaper ones. It’s OK for them to have expensive clothes.”

Help from immediate family members such as siblings also immensely aided single mothers in facing financial difficulties [30]. One respondent expressed her gratitude for having siblings who always helped her financially.

“All my siblings give me financial support.”

“My younger siblings told me to let them know whenever I face problems, they will help.”

## 4. Conclusions

The quality of life of these single mothers has been significantly affected by the pandemic. Their vulnerability towards stress, anxiety, and depression have worsened due to financial issues. Besides the need for emotional and social support, this study found that these single mothers entail financial support. Majority of B40 single mothers expressed their hopes to improve their quality of life by increasing family income so that they can change their lives for the better. Single mothers with school-aged children had to plan on working double as hard to earn more income for their children’s education in the future. Surprisingly, some of them have subscribed to education plan insurance for their children. This study suggests that government support in terms of financial and education planning for children would help improve the quality of life among single mothers in the country.

## Data Availability

Not applicable.

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
