# Peer review of "Perceptions of Quality of Life during the Pandemic: A Case Study on B40 Single Mothers"

_ijerph, 2022, doi:10.3390/ijerph191912219_

Round 1
Reviewer 1 Report
1. The manuscript is to be revised on the methods, specifically sampling method
-What are the population characteristics
-how many are they
-how the authors select the participants for interview relating to method and numbers
2. How the author develop the questionnaire
-what kinds of questions/variables included
-explain process of the development
Reviewer 2 Report
This is a very interesting look into the lives of these single mothers.
It would be helpful if the the conclusion were a bit more in depth and explained the highlights a bit more.
Please explain what B40 means. I'm not from Malaysia and don't know what qualifies someone to be B40.
"devices" is preferred to "gadgets" in the U.S.
Round 2
Reviewer 1 Report
The manuscript has been revised based on the comments suggested.
Author Response
Dear Prof,
Thank you for reviewing our article. The revised article (clean copy) is as attached. Thank you for your kind help.
